# The Dynamic Changes of African Elephant Milk Composition over Lactation

**DOI:** 10.3390/ani10060948

**Published:** 2020-05-30

**Authors:** Sibusiso Kobeni, Gernot Osthoff, Moses Madende, Arnold Hugo, Lisa Marabini

**Affiliations:** 1Department of Microbial, Biochemical and Food Biotechnology, University of the Free State, Bloemfontein 9300, South Africa; Kobenisbu@gmail.com (S.K.); MadendeM@ufs.ac.za (M.M.); hugoa@ufs.ac.za (A.H.); 2AWARE Trust, 16 Southam Road, Greystone Park, Harare, Zimbabwe; lisa@awaretrust.org

**Keywords:** elephant, casein, energy, fatty acid, lactation, milk, oligosaccharide, protein

## Abstract

**Simple Summary:**

The composition of elephant milk differs from all other mammals, as well as between Asian and African elephants. The changes of this milk composition during lactation is also unique. Apart from the major sugar being lactose, sugars also occur as longer chains. With progressed lactation, the content of the lactose decreases, and oligosaccharides become the major sugar component. The content of protein, minerals, and fat also increase during lactation, resulting in an increase in total energy. The fatty acid composition changes during lactation to a high content of saturated acids. Vitamin E occurs at low levels in this milk, and vitamins A, D3, and K occur in trace amounts. The combined data of 14 African elephants over 25 months of lactation are presented. The reported changes may contribute to improving the management strategies of captive African elephants to optimize the nutrition, health, and survival of elephant calves.

**Abstract:**

The combined data of milk composition of 14 African elephants over 25 months of lactation are presented. The milk density was constant during lactation. The total protein content increased with progressing lactation, with caseins as the predominant protein fraction. The total carbohydrates steadily decreased, with the oligosaccharides becoming the major fraction. Lactose and isoglobotriose reached equal levels at mid lactation. The milk fat content increased during lactation, as did the caprylic and capric acids, while the 12 carbon and longer fatty acids decreased. The fatty acid composition of the milk phospholipids fluctuated, and their total saturated fatty acid composition was low compared to the triacylglycerides. The milk ash and content of the major minerals, Na, K, Mg, P, and Ca, increased. Vitamin content was low, Vitamin E occurred in quantifiable amounts, with traces of vitamins A, D3, and K. The energy levels of African elephant milk did not change much in the first ten months of lactation, but they increased thereafter due to the increase in protein and fat content. The overall changes in milk composition appeared to be in two stages: (a) strong changes up to approximately 12 months of lactation and (b) little or no changes thereafter.

## 1. Introduction

In the past few years, the milk of elephants was described to some extent. Compared with milks of all eutherian species, elephant milk is unique with regards to gross and detailed composition. This accounts for Asian (*Elephas maximus*) [1,2] and African elephants (*Loxodonta africana*) [3]. Unique aspects of the fat are (1) a high content of medium chain fatty acids, (2) the compositional changes during lactation, and (3) the low setting and melting temperatures of the milk fat between 4 and 20 °C [3,4,5]. The carbohydrates consist of high amounts of lactose, as is the case with almost all eutherians, but they simultaneously contain high amounts of isoglobotriose, which is a trisaccharide, and oligosaccharides [6,7]. While some of the milk oligosaccharides are shared with other mammals, certain structures are unique to elephants. The whey protein composition seems to be similar to that of other eutherian milks; however, the caseins only consist of β- and κ-caseins, with the β-casein being phosphorylated on one site compared to the multi-phosphorylated equivalents of cow, horse, and human [8,9]. It has also been shown that the milk composition of Asian and African elephants changes drastically with the progression of lactation—more than observed in any other eutherian milk, but almost the same to that of marsupials [2,3]. The drop in carbohydrate content and increase in fat and proteins over the long lactation time of at least two years is similar to that of marsupials [10]. The changes of the macronutrients and minerals in Asian elephant milk with the progression of the lactation were described in detail for four lactations [2]. Similar changes in African elephant milk were shown by a compilation of data from milks from different animals at different lactation stages [3].

Though the composition of African elephant milk had been described to some detail, the whole lactation process was only described by combining data from different individuals, with gaps of data between months 3 and 11 for several parameters—specifically the fatty acid composition. The carbohydrate composition has not been fully described previously, and the quantification of the three fractions was not attempted before. In the current research, the changes in the detailed composition with regards to oligosaccharides, fatty acids, whey, caseins, minerals, vitamins, phospholipids, and energy are described for the first time. In previous work on African elephant milk, it was found that the milk composition of individual elephants varied to a great extent, which complicated the interpretation of changes during lactation [2,4]. We followed a single elephant cow over 19 months, together with shorter periods in three others, and we collected single milk samples of another 10 animals to determine the changes during lactation, as well as the extent of inter-individual differences.

## 2. Materials and Methods

This study complied to the guidelines of the American Society of Mammalogy [11], the Animals Research Ethics of the University of the Free State (UFS-AED2016/0106), and the permission to do research in terms of Section 20 of the Animal diseases act, 1984 (Act No. 35 of 1984) of South Africa (permit reference 12/11/1/4).

### 2.1. Animals and Sample Collection

Milk samples were obtained from three free-roaming African elephants at different lactation stages. Elephant one (Shorty) provided milk for the complete lactation from day one up to 19 months. This elephant lived in Pamuzinda Safari Lodge in the Mashonaland central province, Zimbabwe. In early life, Shorty was used for elephant rides for some years, but this practice was stopped. Interaction with tourists was continued, and this necessitated daily obedience training, taking a total of 1–4 h per day, depending on the number of tourist interactions. During obedience training, a few handfuls of 12% grazer game cubes (Reg. No. V19237 Act 36/1947, EPOL Feeds, South Africa) were fed. For the rest of the day and night, the elephants roamed freely and fed on natural vegetation. The riding and obedience training in early life made Shorty used to human contact and handling, and it provided trust to allow for milk collection without tranquilization and without causing stress. CITES permits (numbers 164,211 and 199,643) were obtained to import the milk to South Africa. Milk samples at 11, 12.2, 13.5, and 14.4 months of lactation were obtained from one additional African elephant (Mussina), and at 21 and 22.5 months of lactation of a second elephant (Shan) that roamed in the Adventures with Elephant’s Reserve, Bela Bela, Limpopo province, South Africa. Shan’s daily routine was similar to that of Shorty, but feeding was supplemented with oats hay and bana grass (a hybrid of *Pennisetum americanum* and *Pennisetum purpureum*). Additional milk samples of various lactation stages were also obtained from one elephant at days 1 and 1.5 months of lactation of the Bloemfontein Zoo (Bloemfontein, South Africa); one elephant at months 13, 14, 17, 19, and 20 of lactation of the Knysna Elephant Ranch (Knysna, South Africa); two elephants at 13 and 24 months of lactation of Sabi Sands Game Reserve (Hazyview, South Africa); three elephants at 12, 14, and 18 months of lactation of the Welgevonden private game reserve (Vaalwater, South Africa); and four elephants at 5, 8, 12, and 18 months of lactation of the Etosha Game Reserve (Namibia). The elephants at the Bloemfontein Zoo was fed alfalfa and a variety of natural grasses, and the one at the Knysna Elephant Ranch followed a similar daily routine as Shorty. The others were free ranging and were tranquilized for routine veterinary procedures. Feeding intervals prior to milk collection were not monitored.

Milk was collected from Shorty at weekly intervals during the first month and two-weekly intervals thereafter. While the baby suckled from one teat, milk was collected from the other by palpation of the teat. This practice was necessary because milk letdown is stimulated by suckling [12]. Teats were milked out as far as possible to avoid possible variations of nutrient content during a milk collection session [13]. The total volumes of collected milk were 5–50 mL. Milk samples were stored frozen and kept at −23 °C during transportation. Once in the laboratory, they were thawed at 39 °C in a warm water bath with gentle swirling, sub-divided in appropriate volumes for individual analytical procedures, and re-frozen. This was done to keep subsequent thawing and re-freezing steps at a minimum. Not all samples collected from Shorty were analyzed. Analyses were carried out on every sample of the first month of lactation in order to monitor the transition from colostrum to milk. Afterwards, one sample of each month was analyzed. Milk samples of a large volume (20–50 mL) were preferred so that all the components could be determined of the same sample and the variation during a milking session could be ruled out. Due to the complexity and expense, the analysis of vitamins and oligosaccharides by Biogel P2 chromatography and fatty acid analysis by gas chromatography were carried out on single samples, but these were followed by a second analysis if large deviations from a compositional tendency were noted. The determination of nitrogen, carbohydrates, and minerals was carried out in duplicate; energy determination was carried in triplicate. The data are presented as averages of two or three.

### 2.2. Determination of Density, Ash and Minerals

African elephant milk density of 10 mL was determined by gravimetric method at 22 °C [14]. A water content of approximately 0.5 g was determined gravimetrically after drying in a forced convection drying oven for 2–3 h at 105 °C and being re-weighed [15]. The dried sample was incinerated at 550 °C for approximately 2 h, and dissolved minerals were analyzed by inductively coupled plasma optical emission spectroscopy (ICP-OES) by the Center of Groundwater Studies, University of the Free State, according to American Public Health Association (APHA) method 3120 B [16].

### 2.3. Vitamin Analysis

Analyses of vitamins A, D_3_, E, and K were carried out by Merieux NutriSciences, Swift Silliker (Pty) Ltd., Claremont, South Africa. Eagle Biosciences Vitamin HPLC assay kits were used: VAE 31-H100 for vitamins A and E [17,18], VD331-H100 for vitamin D_3_, and VK131-H100 for vitamin K [19,20,21]. Reagent preparation, sample preparation, and HPLC analyses were performed as described in the procedures.

### 2.4. Protein Analysis

Two different methods were used. The first was Kjeldahl (AOAC 2005) and Dumas nitrogen combustion analysis method [22] for the analysis of proteins in the African elephant milk. Two milk samples of one elephant (Shorty) at 3.8 and 14.6 months of lactation were analyzed in triplicate. NPN (non-protein nitrogen) and whey proteins were fractionated by selective precipitation with trichloroacetic acid or acidification by hydrochloric acid, followed by centrifugation, according to the method of Igarashi [23], and the nitrogen content of each fraction was determined by the Dumas method. The protein content of the fractions was calculated by the subtraction of NPN and the multiplication of the nitrogen content with a factor of 6.38.

### 2.5. Lipid Analysis

Two different methods, the Röse–Gottlieb (IDF Standards 22B 1987) [24] and the micro-method, described below, were used for the analysis of fats in the African elephant milk. Two milk samples of one elephant (Shorty), at 3.8 and 14.6 months of lactation, were analyzed in triplicate. The quantitative extraction of total fat was performed according to Folch et al. [25] with chloroform and methanol in a ratio of 2:1 (*v*/*v*). The total extractable fat content was determined by weighing and is expressed as g fat/100 g milk. Fatty acids were trans-esterified to form methyl esters with 0.5 N NaOH in methanol and 14% boron trifluoride in methanol [26]. The fatty acid methyl esters (FAME) were quantified using a Varian 430 GC with a flame ionization detector and a fused silica capillary column, Chrompack CPSIL 88 (100 m length, 0.25 mm inside diameter, and 0.2 μm film thickness). The column temperature was 40–230 °C (hold 2 min; 4 °C/min; hold 10 min). The solution of FAME in hexane (1 μL) was injected into the column using a Varian 4800 Autosampler with a split ratio of 100:1. The injection port and detector were both maintained at 250 °C. Hydrogen was used as the carrier gas at 45 psi, and nitrogen was the makeup gas. Chromatograms were recorded using the Galaxy Chromatography Software. Nonadecanoic acid (C19:0) was used as the internal standard after it was established that it was not detected in the samples under study. The identification of FAME sample was made by comparing the relative retention times of FAME peaks from samples with those of standards obtained from Supelco (Supelco 37 Component FAME Mix 47885-U and C18:1c7, C18:2c9t11, C19:0, C22:5).

Phospholipids were separated from the extracted lipid fraction with solid phase extraction using silica-bonded NH2 columns with a 500 mg bed mass, a 3 mL capacity, and a 40 µm mesh size obtained from Agilent (Part No. 12102041, MFG code 204107) according to Bossio and Scow [27]. Quantification of total lipids and separation of different fractions was not attempted. Fatty acid methyl esters (FAME) of the phospholipid fraction were prepared and analyzed as described above.

### 2.6. Carbohydrate Analysis

Saccharides were extracted from milk with 500 µL 25% trichloroacetic acid to 1 mL milk sample and filtered through Nanosep 3K MF Centrifugal Devices (Pall Life Sciences, Michigan, USA) in an Eppendorf centrifuge at 13,000 rpm. The filtrate was subjected to HPLC analysis with a Waters Breeze HPLC system with Biorad Aminex 42C (300 × 7.8 mm) and Water Sugar Pak 1 (300 × 7.8 mm) columns at 84 °C with a differential refractive detector. Elution was by deionized water at a flow rate of 0.6 mL/min. Quantification was done using maltotriose, lactose, glucose, galactose, and isoglobotriose (prepared from African Elephant milk according to Osthoff et al., 2008) as standards.

Mono-, di-, tri-, and oligosaccharides were separated, as described by Osthoff et al. [7]. The elephant milk was extracted with four volumes of chloroform/methanol (2:1, *v*/*v*). The mixture was agitated and the emulsion centrifuged at 6450× *g*, 4 °C for 30 min. The lower chloroform layer and the denatured protein with other unwanted milk contents were separated. The methanol from the upper layer was evaporated, re-dissolved in 2 mL distilled water, and 400 µL samples were loaded on a Bio-Gel P2 (<45 µL, Bio-Rad, Johannesburg, South Africa) column (1.5 × 90 cm, void volume = 70 mL) that was calibrated with glucose, lactose, and dextran. Elution was done by distilled water at a flow rate of 6 mL/h, and fractions were collected every 20 min (2 mL fractions). Aliquots (40 µL) of each fraction were analyzed for hexose with the phenol-sulfuric acid method [28]. Forty µL of each fraction was transferred to a test tube, followed by the addition of 200 µL of a 5% phenol solution and 1 mL of concentrated sulfuric acid. The tubes were allowed to stand for 10 min, shaken, and placed in a water bath at 30 °C for 20 min. The absorbance of each sample was measured at 490 nm.

The quantification of monosaccharides, lactose, and isoglobotriose was carried out by the integration of HPLC chromatogram peaks. For the quantification of oligosaccharides, the areas under peaks of the Bio-Gel P2 chromatograms were calculated. The lactose peak of the latter was assigned the HPLC-obtained concentration of the same sample, and the oligosaccharide content was calculated relative to that of the lactose. The concentrations of monosaccharides, lactose, and isoglobotriose were determined by HPLC and the phenol-sulfuric acid method for comparison. The obtained results were very similar. The concentration of the oligosaccharides was then derived from phenol-sulfuric acid data and re-calculated against the lactose concentration that was obtained by HPLC.

### 2.7. Determination of Energy

The energy content (GE) of 300 µL African elephant milk was measured in triplicate in an adiabatic bomb calorimeter [29]. Milk samples were added to weighed amounts of dried cotton wool (50 mg) in calorimeter cups and dried overnight at 60 °C (to avoid the generation of Maillard products) to a constant weight. After sample combustion, residual fuse wire and residual acid were measured, with corrections applied to calorimetry data. The energy content of the cotton was subtracted from the total energy measurement to obtain milk GE content. The energy content was also calculated using factors derived by Perrin [29] for the energy content of milk fat, carbohydrates, and crude protein using results obtained from reference methods. The following formula was used to calculate the GE: GE = (9.11 kcal/g x% fat + 5.86 kcal/g x% protein + 3.95 kcal/g x% carbohydrate).

### 2.8. Statistical Analysis

Individual scatterplots of time into lactation against individual chemical attributes were constructed for the data of Shorty separately and the data of all animals combined. The data of colostrum were excluded from statistical analysis. Exponential, linear, logarithmic, and polynomial regressions were alternately fitted to each graph to determine the best fit (with respect to the R^2^ value) between time into lactation and chemical attributes. This was done in an attempt to determine the relationship between time into lactation and chemical attributes [30].

## 3. Results and Discussion

### 3.1. Density, Ash and Minerals

The density of the African elephant milk changed over lactation time. During the first eight months of lactation, it varied between 1.0329 and 1.0405 kg/L, which is in the same order as that of bovine, goat, and sheep milk, which have a density of approximately 1.03 kg/L [31,32]. The density decreased to between 1.0205 and 1.0244 kg/L from 10 months to 16 months of lactation and to around 1.0162 kg/L thereafter, which is less than that of the mentioned domestic species. The difference can mainly be ascribed to an increase of total solids, but more specifically to the lactose being exchanged by fat, which is described below.

The ash content of the African elephant colostrum (Figure 1) was 0.30% and changed to 0.17% after four days. It increased to approximately 0.46% around the 12th month of lactation. The increase in ash content during lactation was contrary to that observed in most other species, such as human and mare [33,34,35]. The increase in ash during lactation was mainly due to the simultaneous increase of Na, K, and Mg, which occur as ions, and P and Ca, that are mainly associated with the caseins. The trend lines of data for the milk ash content of Shorty and that of the combined data were very similar.

The African elephant colostrum contained high concentrations of K and Na at 0.120% and 0.109%, respectively, which decreased to 0.015% and 0.071% after three days (Figure 2a). The content of K increased to approximately to 0.186% after 11 months and stabilized thereafter. The Na content remained almost constant up to 12 months of lactation, and it stabilized to around 0.025% thereafter. The Mg showed a similar trend to Na during the lactation, starting at 0.007% and stabilizing at 0.014%.

The trend lines of the data for milk K, Na, and Mg content of Shorty and that of the combined data were very similar. The K content (0.159%) of African elephant milk has been found to be similar to that of cow’s milk, while the Na content (0.042%) has been found to be lower [36]. The observed increases in Na and K were inversely related to the decrease of lactose during lactation, which is discussed below. This may suggest that the changes in osmotic activity was initially caused by salts, and later by lactose, in order to control the milk volume [37]. The Ca and P occurred at low amounts in colostrum, respectively, at 0.034% and 0.019%, and then increased to 0.120% and 0.053% (Figure 2b). This was lower than the Ca and P content recorded for Asian elephants [2] whose Ca content increased from approximately 0.1% at two months to 0.22% at 28 months, and the P content increased from 0.06% to 0.12% at the same stages. The increase of Ca and P throughout lactation was parallel with the increase in casein content. Similar trends have been observed in buffalo milk [38], while the opposite was observed in horse milk [35,36]. Therefore, the current data, and the observations of Abbondanza et al. [2], appear to conclude that the elephant calf has considerable mineral requirements as it doubles in size.

Minor minerals occurred in low yet constant amounts, with Fe below 0.06% and Zn, Fe, Cu, Mn, Cd, and Cr in less than 0.001%. The ash and mineral contents of the milk from Mussina and Shan, respectively, at mid- and late-lactation, were within the same range as determined for Shorty, indicating the extent of inter-individual variation.

### 3.2. Vitamins

Vitamins in African elephant milk were present in minor concentrations (Table 1). The presence, but not the exact content, of vitamin A, D_3_, and K was estimated, because their concentrations were below the analytical limits of the techniques used. The contents were <0.1 mg/kg for vitamin A, <0.3 mg/kg for vitamin E, and <1 µg/kg for vitamins D_3_ and K. The African elephant milk seemed to be void of vitamins during the first month of lactation. Vitamin E was the only vitamin component present throughout the lactation period at 0.36 mg/kg or less. Similar observations have been reported for Asian elephant milk [39], with vitamin E being present throughout lactation [1]. The changes of these vitamins during lactation seemed to be parallel to the milk fat content.

### 3.3. Proteins

For Kjeldahl’s method, the protein content at 3.8 months of lactation was 3.71 ± 0.07%, and at 14.6 months, it was 4.68 ± 0.07%. For the Dumas method, the protein content at 3.8 months of lactation was 3.64 ± 0.19%, and at 14.6 months, it was 5.00 ± 0.14%. A paired t-test gave α > 0.05 (NCSS 11 Statistical Software, 2016). The Dumas nitrogen combustion analysis was selected as the method of choice to analyze the rest of the elephant milk samples. This was based on these results but also on the convenience of use and demand of small volumes of milk compared the Kjeldahl method. It has to be noted that acidic oligosaccharides, which contain N-neuraminic acid, may contribute to the non-protein nitrogen (NPN) [40]; however, the amounts were almost negligible when the protein content was calculated.

The non-protein nitrogen content of the African elephant milk was approximately 0.1% throughout lactation. The total protein content in colostrum was 4.7% and decreased to 2.5%–3.2% after one month of lactation (Figure 3a), similar to that reported by Osthoff et al. [12], and steadily increased afterwards for the rest of the lactation period to around 4%–6.9%. A difference in the concentrations of individual elephant’s milk, due to biological variation, was detectable; however, these were lower than the values reported earlier [41]. Milk from both Shorty and Bloem showed a decrease in protein content from colostrum to 1.5 months of lactation. The trend in the protein levels during lactation was consistent with the previous work of African elephant [4,12,41] and Asian elephant [2]. The trend line of data for the milk protein content of Shorty and that of the combined were very similar. That was due to the high protein content of milk from other elephants at that stage—a result of inter-individual variation [2,4].

The casein content in African elephant colostrum was 2.6%, decreased to approximately 1.5% after five months, and then steadily increased with progressing lactation up to approximately 4% at 25 months postpartum (Figure 3b). The whey content in the colostrum was 2%, followed by fluctuations ranging from approximately 0.1%–2% throughout the lactation period. The trend line of the data for milk casein content of Shorty was polynomial, showing a slight increase towards 20 months of lactation. The trend line of the data for milk whey content of Shorty was logarithmic, showing a stabilization after 10 months of lactation, while that of the combined data was polynomial, showing an increase towards 20 months of lactation. The total protein content in the milk of Mussina and others (specifically from Etosha, Knysna, and Welgevonden) at mid and late lactation was approximately 0.5%–1% higher than that of Shorty, with 0.5%–1% more casein and 0.5%–1% less whey proteins. The current results suggested that the casein was the dominant protein fraction in African elephant milk throughout the lactation cycle. This is different from other monogastric species such as pig, human, cat, and lion, which have high concentrations of whey proteins, especially during early lactation [42,43,44,45,46]. It was, however, similar to cecum fermenters such as the horse [47] and ass [48].

### 3.4. Lipids

For the Röse–Gottlieb method, the fat content at 3.8 months of lactation was 2.44 ± 0.20% and was 2.63 ± 0.37% at 14.6 months. For the micro-method, the fat content at 3.8 months of lactation was 7.63 ± 0.20% and was 7.85 ± 0.46% at 14.6 months. A paired t-test gave α > 0.05 [49]. The micro-method was selected as the method of choice to analyze the rest of the elephant milk samples. This was based on these results, but also on the convenience of use and demand of small volumes of milk, compared the Röse–Gottlieb method.

The milk fat content of the African elephant in colostrum was 2.3% and increased steadily to above 12% after 13 months; it then fluctuated between 8% and 15.5% (Figure 4). The trend lines of data for milk fat content of Shorty and of the combined data were very similar. In a previous report, the fat content of milk from a number of individual elephants seemed to stabilize at approximately 15% [3]. It therefore seems as if fat content is subject to biological variation. The total fat content in the milks of Mussina and others (specifically from Etosha, Knysna, and Welgevonden), at mid and late lactation seemed to support the tendency to fluctuate after 13 months of lactation. Nevertheless, the increase in fat content with advancing lactation appears to be a characteristic of the African elephant [4,12,41]. The increase in milk fat compensated for the decrease in carbohydrates as an energy source. Similar observations have been made on Asian elephant milk, where increases from 7.5% to 15% and 17.5% fat were reported [2].

The fatty acid composition of the African elephant milk fat displayed high amounts of medium-chain fatty acids and low amounts of long-chain and unsaturated fatty acids. This is different from most mammals whose fatty acid composition consists mostly of long-chain and unsaturated fatty acids [5]. The total content of the saturated fatty acids of African elephant milk changed from 72.7% in colostrum to 94.9% at nine months of lactation and around 96% after 19 months of lactation. The dynamic changes of individual fatty acids in African elephant milk during lactation are shown in Figure 5 with selected fatty acids as examples. The C6:0 (caproic acid) was observed at levels between 0.2% and 0.4%. It occurred sporadically before five months of lactation and consistently thereafter. The presence of short-chain fatty acids in milk is usually associated with ruminant species [49] and cecum fermenters such as horse [47] and ass [48]. This indicates that there might be some form of fermentation taking place in the gut of the African elephant. The C8:0 (caprylic acid) and C10:0 (capric acid) contents of the African elephant colostrum were 2.5% and 31.2%, respectively, increased to 8.3% and 59.1% at nine months of lactation, and increased to 10.5% and 62.2% at 19 months (Figure 5a). The contents of C12:0 (lauric acid) and C14:0 (myristic acid) followed a different tendency of changes during lactation. In colostrum, they occurred at 16.9% and 2.8%, respectively, and increased to around 27% and 4% after one month. The amounts continued to decrease to 18.9% and 1.4% at nine months and remained constant for the rest of lactation. The contents of C16:0(palmitic acid) and 18:1 (oleic acid) acids followed a decreasing trend, with 15.2% and 18.4% in colostrum, respectively, before decreasing to 3.6% and 4.34% at nine months of lactation and 1.95% and 0.1% at 22 months (Figure 5b). The trend lines of data for milk fatty acid content of Shorty and of the combined data were very similar.

A low amount of fatty acids of uneven carbon number were observed in small amounts and showed tendencies of change during lactation. The changes were similar to that of the even numbered ones of approximately the same length. The C11:0 occurred at less than 1% and increased to above 2%, the C15:0 and C21:0 decreased from 0.2% to less than 0.01%, and the C17:0 decreased from 2% to less than 0.01%. The same accounted for C16:1c9 (palmitoleic acid) and C18:3c9,12,15 (α-linolenic acid) long-chain unsaturated fatty acids, which decreased from 1% to less than 0.1%, while C20:2c11,14 (eicosadienoic acid), C20:3c11,14,17 (eicosatrienoic acid), and C22:1c13 (erucic acid) were detected in trace amount only up to 11 months of lactation. No C20:0 (arachidic acid), C22:0 (behenic acid), or C24:0 (lignoceric acid) were detected. The fatty acid content of the milk from Mussina and Shan supported the above findings at mid and late lactation, respectively. However, milk from Mussina, which was collected from months 11–14, contained higher amounts of unsaturated and lower saturated fatty acids. It might be argued that the oats in the supplementary food could have been the source of these fatty acids. However, the same was not observed for the milk fat of Shan at 21 and 22.5 months of lactation, even though Shan received the same fodder. A detailed inspection of the fatty acid data showed similar variations in milk fatty acid between elephants of the same reserve (results not shown separately).

In general, the fatty acids of 10 carbons in length and shorter increased during lactation, while those of 16 carbons and longer decreased, with 12:0 and 14:0 expressing the least change. All these changes seemed to occur in two phases; drastic changes from day zero to nine months and slow changes thereafter. Though the phases of change were noted in a previous report [3], a lack of data in the early to mid-lactation range made accurate description impossible. It is to be noted that the pattern of changes in milk fatty acids observed here differed from that reported by McCullagh andWiddowson [4], who reported amounts of 50% of C10:0 at three months. The current results, and the earlier results of Osthoff [3], showed that, in spite of large biological variation, this level was only reached after at least seven months of lactation. In the current work, the stage of lactation was recorded according to birth date of calves, while McCullagh and Widdowson [4] based the ages of the calves on the stage of dental development [50]. The latter resulted in an underestimation of the stage of lactation and therefore an inaccurate description of the changes of milk fatty acids during lactation.

As was described for the general changes of milk composition above, the changes in fatty acid composition during lactation seemed to be independent of the seasonality of nutrition.

The amounts of C8:0 in African elephant milk are significantly higher than that of the cow, mare, and human [44,51]. The only species that display high concentrations of short/medium chain fatty acids known thus far are the rabbit (C8:0–C10:0) and both Indian and white rhinoceroses (C10:0–C12:0) [7,52,53,54]. The described tendency of changes towards shorter milk fatty acids during lactation has not been documented for any other mammal. The only reason for such a change may lie in the energy provision of the fat. A comparison of the energetic balance of the total oxidation of the fatty acids between milk fat from cow and elephant has shown that the elephant may provide approximately 6% less energy at late lactation. This is compensated for by the synthesis of medium length fatty acids that may provide a saving of energy to the elephant cow [55]. The energy density of elephant milk is therefore increased by the fat content [2] and not by the change in fatty acid composition.

The fatty acid composition of the phospholipids of African elephant milk showed extensive variation during lactation, with no tendencies of increase or decrease. For example, two extreme compositions in subsequent sampling times of Shorty, together with the averages during lactation, are presented in Table 2. The data of elephants Mussina and Shan are also shown. The fatty acid composition of the phospholipids did not resemble that of the triacylglycerides. The total saturated fatty acids in the phospholipids were low compared to the triacylglycerides, at 50%–73% vs. 72%–96%, respectively. The short-chain fatty acids (C4:0 and C6:0) were absent. Of the medium-chain fatty acids, C8:0 was present only in four milk samples of Shorty at 0.3%–33%, while C10:0 and C12:0, respectively, occurred at 10%–35%, and 1%–4%. The C16:0 and C18:1c9 acids occurred at 3%–28% and 13%–37%, respectively. The C10:0 and C12:0 acids occurred in much lower amounts, while the C16:0, C18:0 (stearic acid), and C18:1c9 acids occurred in much higher amounts than in triglycerides. Fatty acids of uneven carbon numbers, C11:0 (hendecanoic acid), C15:0 (pentadecylic acid), C17:0 (margaric acid), and C21:0 (heneicosanoic acid) occurred at the same low amounts as in the triglycerides, and traces of C19:0 (nonadecanoic acid) and C23:0 (tricosanoic acid) were also detected. However, other than in the triglycerides, mono- and polyunsaturated fatty acids of 16–24 carbons in length were detected at amounts below 1% and throughout lactation. In general, the fatty acid composition of the phospholipids did not resemble that of the triacylglycerides, but it was comparable with bovine, pig, and human milk phospholipids because it contained high amounts of long-chain fatty acids [56,57,58,59].

### 3.5. Carbohydrates

The total carbohydrate content of African elephant colostrum was between 3% and 5%. It then increased to above 11% after the first month, where it remained for another month (Figure 6). After the second month postpartum, a steady decrease in total carbohydrate content followed for the rest of the lactation period to around 6%. Milk from both Shorty and Bloem showed an increase in all the saccharides from colostrum to 1.5 months of lactation. The total carbohydrate levels of milk form Mussina and others were within the range of that of Shorty. The trend lines of data for the milk carbohydrate content of Shorty and of the combined data were very similar. Monosaccharides such as glucose and fucose were present in constant proportions below 0.2% throughout the lactation period.

Lactose was the dominant carbohydrate in the early lactation of the African elephants under study, increasing from 1.9% in colostrum to 5.0% after three days, followed by a constant decrease to approximately 2%. Similar observations were made in earlier reports [4,12,41] and were also observed in the milk of reindeer, moose, and black-tailed deer [60,61,62]. A similar pattern of diminishing lactose content was also reported for Asian elephant milk [2], though at half the amounts. Different methods of analysis and quantification were employed in the two studies; as such, only the trend, not the carbohydrate amounts, can be compared.

The results of quantification of the monosaccharides, lactose, and isoglobotriose obtained by HPLC and the phenol-sulfuric acid method were very similar. The accurate quantification of acidic oligosaccharides was reported to be best carried out by chemical labeling and separation, followed by the quantification of individual oligosaccharides by HPLC [63]. A quantification method other than the phenol-sulfuric acid method of the Biogel P2 chromatography peaks has not yet been described. Confidence in the method used here may be provided by the fact that all the milk components of the milk samples from the African elephants reported here added up to approximately 100%, and the average for all samples was 100.6 ± 9.3%.

At day zero, the oligosaccharide content of the African elephant milk was 2.1% (Figure 7a), and it increased to 4.3% within the first month of lactation. This was followed by a steady decrease to around 3.3% over the rest of the lactation period. The quantities of oligosaccharides in the milk of Mussina and Shan, at mid and late lactation, respectively, were approximately 0.5% lower, but they seemed to support this trend. The oligosaccharides became the major carbohydrates from approximately the fourth month of lactation. The trend line of data for the milk oligosaccharide content of Shorty was polynomial, while that of the combined data showed a decrease from early lactation and stabilization from 12 months onward. The observed increase in oligosaccharide content is in agreement with earlier reports around a similar lactation time [12,41]. High levels of milk oligosaccharides are usually linked with monotremes, marsupials, and bears [6,64,65].

The isoglobotriose content increased from 0.8% in colostrum to 2.2% at 12 months of lactation; it subsequently decreased to 1.5% at 19 months (Figure 7b). It became the major carbohydrate from approximately the fourth month of lactation. The quantities of isoglobotriose in the milk of Mussina at mid lactation was approximately 0.5% higher than that of Shorty. The trend lines of data for the milk isoglobotriose content of Shorty and of the combined data were very similar. Isoglobotriose has also been observed in other mammalian species such as the Asian elephant, polar bear, coati, and giant panda [6,66,67,68].

A similar trend of change in saccharide composition has also been observed in marsupial milk, where oligosaccharides become main carbohydrates over the lactation period [65]. In human milk, the opposite trend was observed with a significant increase of lactose and a concomitant decrease of oligosaccharides with advancing lactation [63,69]. The presence of high oligosaccharide levels in African elephant milk may be due to different types of galactosyltransferases [70]. It has not yet been established how these transferases are regulated to allow for the simultaneous synthesis of high amounts of lactose, isoglobotriose, and oligosaccharides.

### 3.6. Energy

The measured GE levels of the African elephant milk remained at around 25 kCal/100 g to approximately nine months of lactation and increased thereafter to around 80 kCal/100 g (Figure 8). This increase was due to the increase of, mainly, the protein and fat contents. The trend lines of the data for milk gross GE of Shorty and of the combined data were very similar. The calculated GE of the African elephant milk was twice as high as the experimentally measured GE, and it was approximately half of the GE reported by Abbondanza et al. [2]. These researchers also used the calculation formula of Perrin [29] but cautioned that the calculated energy values were most probably an over-estimation. The great deviation between calorimetric GE and calculated GE may have been due to the components of African elephant milk not being the same as those milks used by Perrin [29]. When Perrin devised the calculation formula, separate mathematical compensations were made to obtain calculated values that were similar to the experimental calorific values. Moreover, the component content of the milks of the species included by Perrin [29], cow, ewe, goat, and pig did not differ much from each other. At the early months of lactation, elephant milk did not differ too much from most of the mammals in the publication with regard to the content of fat, protein, and carbohydrates. Hence, the difference in the calculated and experimentally measured values was not high. However, after 10 months of lactation, the calculated values were twice as high as the experimental ones, which may have been due to the high content of all components, as well as the difference in the detailed composition of each of the components.

With regard to the detailed composition, the fatty acid and carbohydrate composition of African elephant milk differs substantially from other mammalian species. Perrin [29] based their calculations on the species of which the milk composition consisted of fat that was very similar to cow’s milk and contained long-chain fatty acids. That means that there is more fatty acid per gram fat than glycerol, while in elephant milk, and due to the high amount of medium-chain fatty acids, there is more glycerol per gram fat—approximately 1.2 times. It is possible that this may have led to an over-estimation of the calculated GE of fat in our study, but this has to be determined experimentally. The greatest contributor to the high calculated energy values may have been the carbohydrates. Perrin’s milk carbohydrates were all assumed to be lactose, while in elephant milk lactose is only one-third of the total carbohydrates, and the energy values of isoglobotriose and oligosaccharides are not known. Therefore, this concludes that the formula used to calculate the values of energy only applies to milk that is similar to that of cows or other ruminants, and adjustments would be needed for other mammalian species [29,71]

## 4. Conclusions

In general, the colostrum composition of African elephant differs from that of the early lactation milk, which is also seen in most other mammals, e.g., the cat [42] and pig [43]. Here, it was found to consist of a high ash content, specifically Na and K, but low Ca and P contents, a high protein content, and low carbohydrates and fat. The milk composition changed throughout lactation, seemingly in two stages. While the protein content seemed to change continuously during lactation, the changes of ash, Ca, P, fat, fatty acids, and lactose were drastic up to approximately 12 months, followed by a stage of little changes. The large difference of inter-individual milk components may have influenced data interpretation. This was pointed out by the statistical analysis of data from total protein, whey, oligosaccharide, and isoglobotriose contents.

It may be questioned whether the changes in milk composition were due to nutritional changes over season. The data of McCullagh and Widdowson [4], and some of the accumulated data from individual elephants [3], were from milk collected at the same time, though the lactation stage differed. In these cases, the nutrition was constant, while the stage of lactation changed. In other examples from Osthoff [3], the individual elephants were at similar lactation stages, but the milk was collected at different times of the year or at different environments. In all these cases, the milk composition was dependent on lactation stage and independent of the seasonality of nutrition.

The results of the milk from one additional elephant at the beginning of lactation and two at later lactation, together with accumulated data from other individuals [3], showed inter-individual variation. The current data, as well as the accumulated data, support changes of milk composition in two stages. This is in contrast with the Asian elephant milk, where a continuous change of all components has been reported [2,72].

## Figures and Tables

**Figure 1 animals-10-00948-f001:**
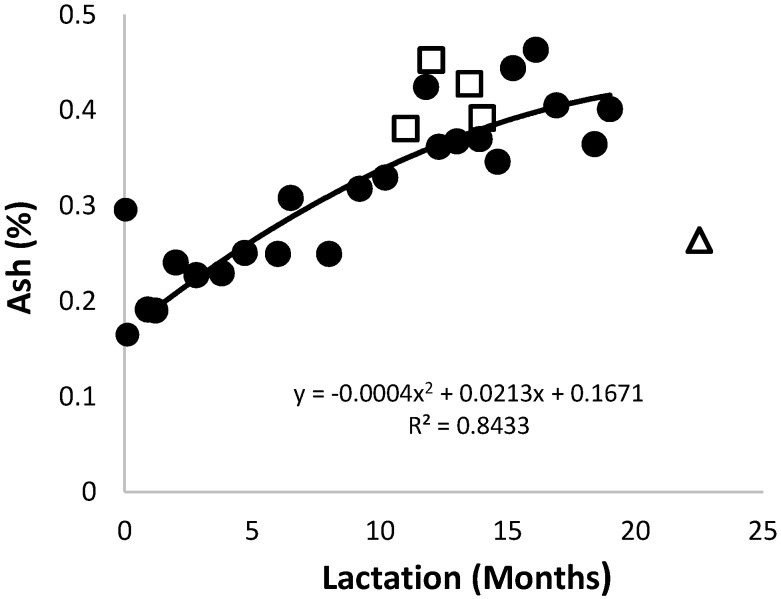
Changes of the ash content in milk of three African elephants during lactation (Shorty (●), Mussina (□), and Shan (Δ)). The trend line of combined data of all animals was y = 0.0122x + 0.1671 and R^2^ = 0.8297.

**Figure 2 animals-10-00948-f002:**
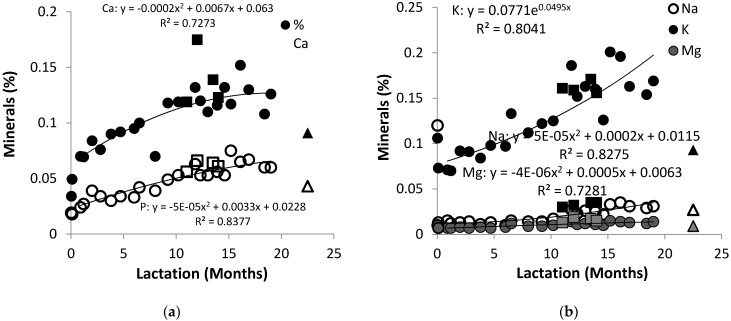
Changes of (**a**) Ca and P in milk of three African elephants during lactation (Shorty (○●), Mussina (□■), and Shan (Δ▲)) and (**b**) Na, K, and Mg (Shorty (○●●), Mussina (□■■), and Shan (Δ▲▲)). The trend lines for combined data of all animals were y = −0.0005x^2^ + 0.0314x + 0.053 and R^2^ = 0.7023 for Ca, y = −0.0004x^2^ + 0.0589x + 0.0228 and R^2^ = 0.8213 for P, y = 0.0071x + 0.073 and R^2^ = 0.7981 for K, y = 0. 5 x^2^ + 0.0007x + 0.0115 and R^2^ = 0.8151 for Na, and y = 0.000005x^2^ + 0.0005x + 0.0063 and R^2^ = 0.7182.

**Figure 3 animals-10-00948-f003:**
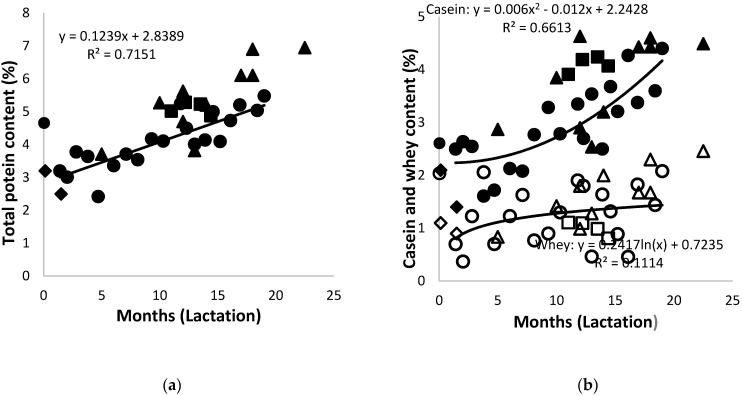
(**a**) Changes of the total protein content lactation (Shorty (●), Bloem (♦) Mussina (■), and others (▲)) and (**b**) of casein (solid symbols) and whey proteins (open symbols) in milk of African elephants during lactation (Shorty (●○), Bloem (♦◊) Mussina (■□), and others (▲Δ)). The trend lines for combined data of all elephants were y = 0.0027x^2^ + 0.1076x + 2.9055 and R^2^ = 0.7153 for total protein, y = 0.126x + 1.8339 and R^2^ = 0.6089 for casein, and y = 0.0034x^2^ − 0.0243 + 1.0522; and R^2^ = 0.2815 for whey.

**Figure 4 animals-10-00948-f004:**
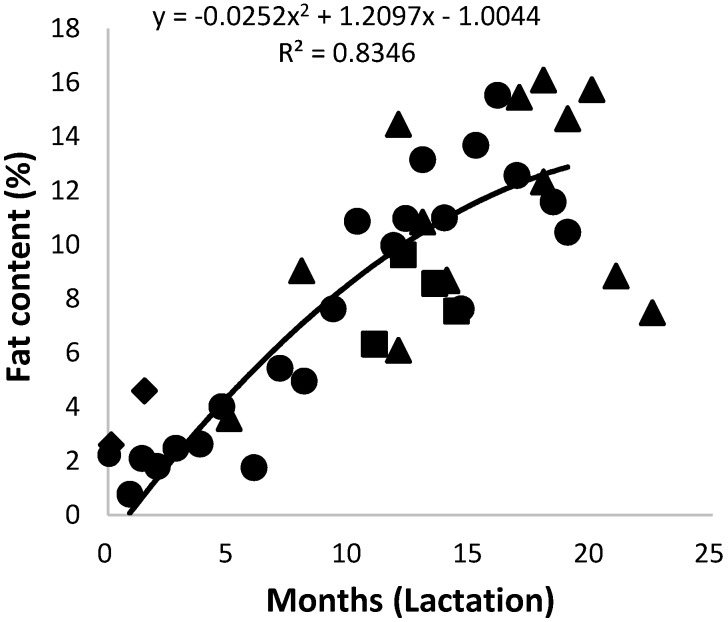
Changes of the total fat content in milk of African elephants during lactation (Shorty (●), Bloem (♦) Mussina (■), and others (▲)). The trend line of combined data of all elephants was y = −0.0242x^2^ + 1.1098x − 0.0308 and R^2^ = 0.6832.

**Figure 5 animals-10-00948-f005:**
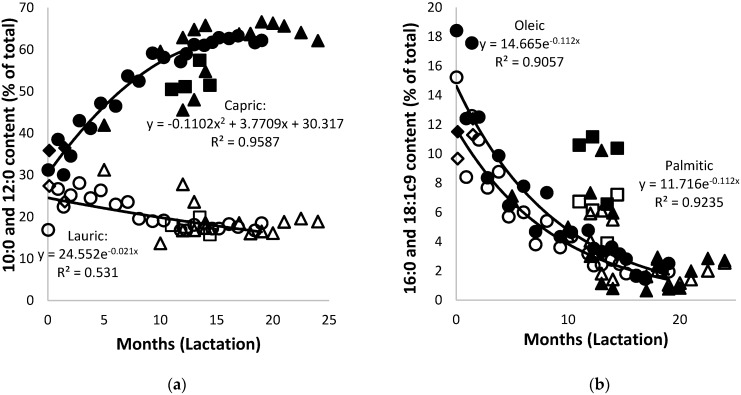
(**a**) Changes of capric (solid symbols) and lauric acids (open symbols) and (**b**) of palmitic (solid symbols) and oleic acids (open symbols) in milk of African elephants during lactation (Shorty (●○), Bloem (♦◊) Mussina (■□), and others (▲Δ)). Trend lines for combined data of all elephants were y = −0.0617x^2^ + 2.802x + 32.557 and R^2^ = 0.9421 for C10:0, y = 0.0203x^2^ − 0.8453x + 26.226 and R^2^ = 0.4451 for C12:0, y = 0.0211x^2^ − 0.9033x + 11.591 and R^2^ = 0.7818 for 16:0, and y = 0.0217x^2^ − 1.018x + 13.997 and R^2^ = 0.6586 for C18:1c9.

**Figure 6 animals-10-00948-f006:**
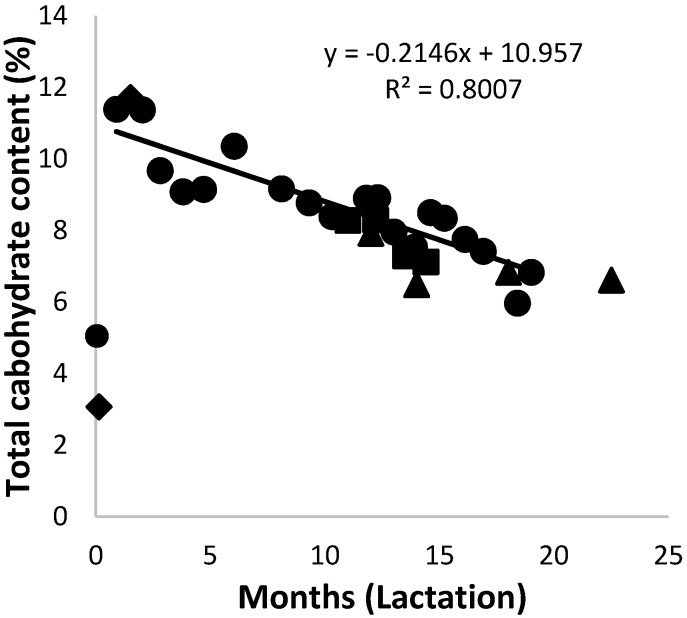
Changes of the total carbohydrates in milk of African elephants during lactation (Shorty (●), Bloem (♦) Mussina (■), and others (▲)). The trend line for combined data of all animals was y = 0.0051x^2^ + −0.3381x +11.444 and R^2^ = 0.8214.

**Figure 7 animals-10-00948-f007:**
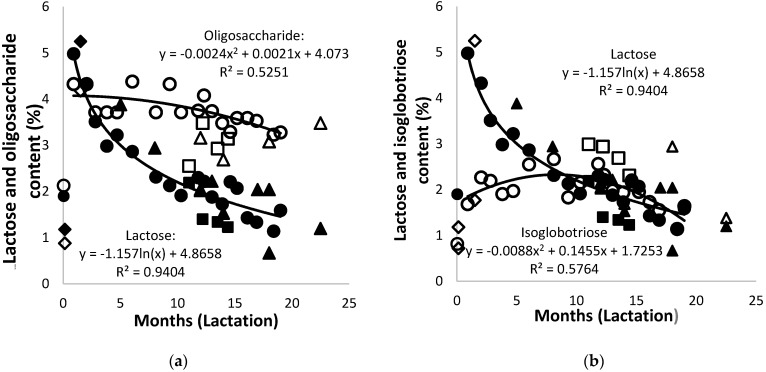
(**a**) Changes of lactose (solid symbols) and oligosaccharides (open symbols) and (**b**) of lactose (solid symbols) and isoglobotriose (open symbols) in the milk of African elephants during lactation (Shorty (●○), Bloem (♦◊) Mussina (■□), and others (▲Δ)). Trend lines for combined data of all animals were y = −1.313ln(x) + 5.1479 and R^2^ = 0.9146 for lactose, y = 0.003x^2^ + 0.1128x + 4.3682 and R^2^ = 0.3998 for oligosaccharides, and y = 0.0068x^2^ + 0.1292x + 1.7396 and R^2^ = 0.2966 for isoglobotriose.

**Figure 8 animals-10-00948-f008:**
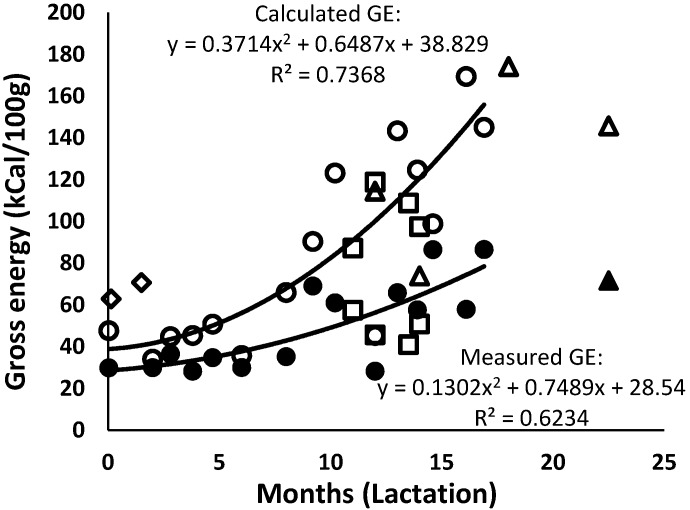
Changes in measured (solid symbols) and calculated (open symbols) energy in milk of African elephants during lactation (Shorty (●○), Bloem (◊) Mussina (■□), and others (▲Δ)). The trend lines for the combined data were y = 27.946e^0.005x^ and R^2^ = 0.5621 for the measured energy content (GE) and y = 0.1079x^2^ + 3.5884x + 41.778 and R^2^ = 0.6571 for the calculated GE.

**Table 1 animals-10-00948-t001:** Vitamin content of African elephant milk during lactation.

Lactation Time (Months)	Vit A (mg/kg)	Vit D_3_ (µg/kg)	Vit E (mg/kg)	Vit K (µg/kg)
0.03	0.1	ND	0.36	ND
0.1	ND	ND	0	ND
0.9	ND	ND	0	ND
1.4	ND	ND	<0.3	<1.0
2.03	ND	<1.0	<0.3	<1.0
2.8	ND	ND	0.33	ND
3.8	ND	ND	0.31	ND
4.7	ND	<1.0	<0.3	<1.0
6.03	<0.1	ND	0.3	<1.0
7.1	<0.1	ND	<0.3	<1.0
8.1	<0.1	ND	<0.3	<1.0
9.3	<0.1	ND	<0.3	<1.0

**Table 2 animals-10-00948-t002:** Milk total phospholipid fatty acid content of milk from three African elephant cows during lactation.

Time of Lactation (Months)		Shorty 12.3	Shorty 13.0	Shorty Average (*n* = 16)	Mussina Average (*n* = 4)	Shan Average (*n* = 2)
Fatty Acid Composition (%):						
Common Name:	Abbreviation:					
Butyric	C4:0	0.0	0.0	0.00 ± 0.00	0.00 ± 0.00	0.00
Caproic	C6:0	0.0	0.0	0.00 ± 0.00	0.00 ± 0.00	0.00
Caprylic	C8:0	13.3	0.0	1.96 ± 4.49	0.00 ± 3.93	0.00
Capric	C10:0	54.7	6.9	20.17 ± 14.55	6.30 ± 13.61	6.34
Hendecanoic	C11:0	1.0	4.5	1.05 ± 1.13	1.56 ± 1.00	1.43
Lauric	C12:0	7.4	1.5	7.68 ± 4.60	1.93 ± 4.55	5.36
Tridecanoic	C13:0	0.0	0.0	0.03 ± 0.07	0.00 ± 0.06	0.00
Myristic	C14:0	0.5	1.2	1.85 ± 0.98	1.24 ± 0.88	2.85
Myristoleic	C14:1c9	0.0	0.0	0.04 ± 0.08	0.51 ± 0.24	0.08
Pentadecylic	C15:0	0.1	1.7	0.24 ± 0.42	0.00 ± 0.37	0.16
Pentadecenoic	C15:1c10	0.0	0.0	0.00 ± 0.00	0.00 ± 0.00	0.00
Palmitic	C16:0	3.9	20.3	15.91 ± 6.74	21.46 ± 6.56	23.42
Palmitoleic	C16:1c9	0.3	1.8	0.81 ± 0.70	0.90 ± 0.68	1.24
Margaric	C17:0	0.3	0.5	0.55 ± 0.35	0.50 ± 0.31	0.63
Heptadecenoic	C17:1c10	0.4	0.1	0.41 ± 0.36	0.98 ± 0.42	0.24
Stearic acid	C18:0	3.6	10.9	13.13 ± 7.20	18.64 ± 7.00	12.74
Elaidic	C18:1t9	0.1	0.5	0.18 ± 0.24	0.13 ± 0.22	0.08
Oleic	C18:1c9	11.1	30.0	24.12 ± 6.73	30.50 ± 7.99	30.12
Vaccenic	C18:1c7	0.1	2.7	0.80 ± 0.91	1.19 ± 0.91	1.06
Linolelaidic	C18:2t9,12 (n−6)	0.0	0.6	0.29 ± 0.41	1.30 ± 0.63	0.00
Linoleic	C18:2c9,12 (n−6)	1.0	9.0	2.89 ± 2.19	3.19 ± 2.69	6.37
Conjugated linoleic acid (CLA)	C18:2c9t11 (n−6) (CLA)	0.0	0.0	0.03 ± 0.05	0.14 ± 0.07	0.00
Conjugated linoleic acid (CLA)	C18:2t10,c12(n−6)(CLA)	0.0	0.3	0.04 ± 0.08	0.91 ± 0.62	0.02
γ-Linolenic	C18:3c6,9,12 (n−3)	0.0	0.4	0.18 ± 0.32	1.35 ± 0.63	0.32
α-Linolenic	C18:3c9,12,15 (n−3)	0.8	0.1	0.73 ± 0.58	0.76 ± 0.54	0.69
Nonadecanoic	C19:0	0.0	0.0	0.50 ± 0.89	0.17 ± 0.78	1.71
Arachidic	C20:0	0.1	0.0	0.26 ± 0.17	0.19 ± 0.91	0.19
Eicosenoic	C20:1c11	0.0	0.2	0.23 ± 0.35	0.00 ± 0.31	0.06
Eicosadienoic	C20:2c11,14 (n−6)	0.0	0.4	1.03 ± 3.20	0.50 ± 2.76	0.47
Eicosatrienoic	C20:3c8,11,14 (n−6)	0.1	0.2	0.30 ± 0.25	0.70 ± 0.41	0.03
Eicosatrienoic	C20:3c11,14,17 (n−3)	0.1	0.0	0.05 ± 0.05	0.31 ± 0.19	0.09
Arachidonic	C20:4c5,8,11,14 (n−6)	0.3	2.8	0.45 ± 0.64	0.48 ± 0.57	1.84
Eicosapentaenoic	C20:5c5,8,11,14,17 (n−3)	0.0	0.6	0.15 ± 0.20	0.09 ± 0.17	0.03
Heneicosanoic	C21:0	0.0	0.1	0.04 ± 0.04	0.32 ± 0.20	0.00
Behenic	C22:0	0.0	0.1	0.14 ± 0.24	0.16 ± 0.22	0.15
Erucic	C22:1c13	0.1	0.1	0.43 ± 0.81	0.38 ± 0.70	0.27
Docosadienoic	C22:2c13,16 (n−6)	0.0	0.1	0.06 ± 0.09	0.20 ± 0.16	0.01
Docosapentaenoic	C22:5c7,10,13,16,19 (n−3)	0.01	0.92	0.33 ± 0.35	1.39 ± 0.64	0.41
Docosahexanoic	C22:6c4,7,10,13,16,19 (n−3)	0.0	0.0	0.05 ± 0.07	0.05 ± 0.06	0.06
Tricosanoic	C23:0	0.5	1.4	1.30 ± 1.80	0.77 ± 1.61	0.42
Lignoceric	C24:0	0.0	0.0	0.07 ± 0.08	0.22 ± 0.09	0.09
Nervonic	C24:1c15	0.1	0.2	0.44 ±1.05	0.58 ± 0.91	1.05
**Fatty Acid Ratios:**						
SFA (%)		85.4	49.0	65.97 ± 9.10	53.45 ± 15.14	55.47
MUFA (%)		12.3	35.7	27.45 ± 7.56	35.18 ± 8.86	34.20
PUFA (%)		2.3	15.3	6.58 ± 3.85	11.37 ± 3.93	10.33
n−6 (%)		1.4	14.2	5.36 ± 4.05	8.67 ± 3.84	9.08
n−3 (%)		0.92	1.63	1.30 ± 0.72	2.60 ± 0.97	1.28

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
