# Peer review of "The Dynamic Changes of African Elephant Milk Composition over Lactation"

_animals, 2020, doi:10.3390/ani10060948_

Round 1

Reviewer 1 Report

Dear authors, I appreciated the explanation of the authors about the challenges to collect elephant milk in real life. Very very interesting. I totally understand the constrains of sampling a wild species and appreciate the effort in combining this data into a manuscript. I also appreciated the changes that were substantial, but made the difference. By combining results and discussion, the text is easy to read, and interesting. Well-done! It was with pleasure that I have read this manuscript now, and I am pleased to accept it for publication.

Reviewer 2 Report

Dear Authors,

I reread the paper and found that many comments were considered. But I think the Regression equations on the figures are not necessary, especially on figure 2b and figure 3b.

Good luck

This manuscript is a resubmission of an earlier submission. The following is a list of the peer review reports and author responses from that submission.

Round 1

Reviewer 1 Report

Review for Animal Journal – Kobeni et al. (2020)

Comments to the authors:

Major comments:

The authors report “The dynamic changes of African elephant milk composition over lactation”. This subject is interesting because it tackles a wild species, endangered by Mankind, and in desperate need of conservation. Furthermore, a study on detailed milk composition is not commonly done in commercial species, such as bovine. Therefore, this study on elephants is important and needed!

While the subject is quite interesting, there are some issues in the paper that prevent it’s acceptance in the current format. These issues are: 1) a confusion between Results and Discussion section, 2) some missing parts of the material and methods, that can be found in the Results section, and 3) some confusion of terms that are not used in the field of Milk and milk composition, because of the distinct origins of the components in milk (further commented in the detailed comments).

The issues 1 and 2 reflect a lack of structure in the manuscript. My suggestions to the authors are to consider combining Results and Discussion Section, use sub-headers to distinguish between the different topics in this new combined Results and Discussion, as well as thoroughly rethink the material and methods because with the use of sub-headers, the authors could create a separation between animals, analyses of milk  (or phenotypes), statistical analyses, etc…. By structuring the paper with the aid of sub-headers, the paper improves considerably.

In general, the paper is well written, despite a few mistakes here and there.

Therefore, I am recommending this paper for a major revision.

Detailed comments:

Abstracts:

Line 3: remove dot at the end of the title

Line 14: consider re-arranging this sentence to line 13, after mammals, and changing it to “(…) mammals, and differs between Asian and African elephants. The changes…”

Line 18: please consider changing this sentence to: “(…), and the fatty acids composition change.”

Line 19: please combine sentence 19 and 20 as follows: “The milk composition becomes stable at approximately 12 months lactation, resulting in an increase in total energy.

Line 29: The milk fat content increased over lactation or at mid-lactation?

Line 35: Consider changing this sentence because you specify two stages, but you only mention one. Possibly what you are trying to say is: “The changes in milk composition appear to be in two stages : a) no major changes from zero to 10 months of lactation, followed by b) strong changes up to approximately 12 months of lactation, after which most of the nutrients stabilize.”

Keywords: Please remove nutrition.

Introduction:

Line 42: add a comma after (…) few years, the (…)

Line 42: Please change compared to and use compared with,

Line 43: regards to gross and detailed milk composition.

Line 45: Unique aspects of the fat are 1) a high content of medium chain fatty acids, 2) the changes (of what?) over lactation periods, and 3) the setting and melting properties of fat, from as low as 4°C  up to XX°C [3-5].

Line 47: (…)of isoglobotriose, which is a trisaccharide,(…)

Line 51: (…)with the β-casein being phosphorylated on one site,

Line 53-54: please change “nutrient composition of the milk of both elephant species changes” to “the milk composition of Asian and African elephants changes (….)

Line 58: Please change:  with the progression of the lactation

Line 59 : please change: from milks from

Line 62: please change: (…) described by combining data (…)

Line 64: this sentence (, while the changes of the proteins were only needed updating.) does not read correctly. Please consider re-writing or removing.

Line 65: consider changing: “was previously described fragmentary,” to “was not fully described previously,

Line 66: with regards to

Line 68: We followed a single elephant cow  over 19 months, together with shorter periods in three others, and collected single milk samples of another 11 animals, to determine the extent of inter-individual differences.

Material and Methods:

Line 78-79: the authors state that milk samples were collected from 3 elephants, and elephant one is thoroughly described. What about Elephants two and tree? Please add details about elephants 2 and 3 in this paragraph. I realize that at line 89 details have been given for Mussina. But there are not enough details as compared with Short. I suggest the authors to continue their style in this paragraph and add “Elephant two was …..”. Then, further on, “Finally, elephant 3 was….”.

Line 88: Cities permit (…)?

Line 91: please remove above, because it is not needed.

Line 92: How many additional samples were included in the study? From how many animals? Was the daily routine also similar to Shorty’s? Could the authors add day ranges for early, mid and late lactation, instead of “various lactation stages”?

Line 111: Please change to duplicates and triplicates (throughout the paper)

Lines 115 and 116: please change to “(..) determined by gravimetry …”

Line 129 and 138 : Please change to “Elution was obtained by adding”

Line 140: Please write 40 in full “Forty “, and replace was by were.

Line 141: please add after followed by an addition of

Line 150 and Line 158: Please specify which samples, because “the same sample ( and all other samples)” is unclear.

Line 154: please add was before determined.

Line 155 : please change to  “(…) by the subtraction of NPN and the multiplication of the (…)”

Line 156:please change the “for the determination of the fat content of (…)” to “used to determine the fat content of the (…)”

Line 170 : It is nonadecanoic, not Nondecanoic. Please change.

Line 180: add (GE) after energy content. It will be easier for the reader.

Add to the material and methods that the comparison between the samples was done based on a paired T-Test, and that the significance level was  at alpha = 0.05 . As a consequence, please remove the lines tackling this issue from the Results section (Lines 199,and  224).

Results:

The comparison between the elephants and/or two distinct lactation stage is difficult to read. There are too many respectively. My first suggestion to the authors is to re-organize all the comparisons across the paper. As an example, from lines 195-203, it will be easier to read the following paragraph: “

Two different methods were used, Kjeldahl and Dumas nitrogen combustion analysis, for the analysis of proteins in African elephant milk. Milk samples from two elephants were compared. These samples were at 3.8 and at 14.6 months of lactation, and they were analyzed as triplicates. For Kjeldahl’s method, the protein content at 3.8 months of lactation was 3.71 ± 0.07%, and at 14.6 months was 4.68 ± 0.07%. For the Dumas’s method, the protein content at 3.8 months of lactation was 3.64 ± 0.19% and at 14.6 months was 5.00 ± 0.14%. The Dumas nitrogen combustion analysis was selected as method of choice to analyze the rest of the elephant milk samples. It has to be noted that acidic oligosaccharides, which  contain N-neuraminic acid, may contribute to the non-protein nitrogen (NPN) [30], however, the amounts are almost  negligible when the protein content is calculated.”

However, it is good to realize that 1) repetition is needed to be precise and consistent, and 2) there is a confusion between Results and Material and Methods. For instance, the highlighted text in italic blue belongs to Material and Methods (and also lines: 221-222;304-305;306-307;etc…) . The part highlighted in underlined green belongs to the Discussion section(and also lines: 218-220; 228-229;234-235;254-255;262-263,266-267, 308-313;etc) . The issue is seen throughout the Results section, and this is confusing.

My second major suggestion to the authors is to write the Results and Discussion section together. It will solve most of the issues.

I also have a question about this selection of methods (lines 200,and 225) : what is the criteria? Please add this criteria in the Material and Methods.

Please add to the material and methods the software used to create all your figures.

Lines 217 and line 228: Who are the others? Please specify!

Line 221: Who are the two elephants? Please specify!

Line 239: In table 1, fatty acids are written as C6:0, C10:0, C12:0. This is, by the way, the correct way of writing about Fatty acids. Please fix by adding a C in front of all fatty acids described in the results,and discussion sections.

Lines 242-245: This sentence is too long, and not clear at all. Please re-write.

Figure 3: Mussina received oats. Could this be the reason why the fatty acids content is different from the other elephants?

Figure 3: Were the elephants defined as others grouped in this study? It should be clear in the material and methods.

Line 267: Please change “as a demonstration” with “for example”

Discussion:

Please reconsider the use of the term nutrient. I realize that it is a correct term, but most people working with milk, milk composition, fatty acid composition, vitamins, ashes and minerals , do not use the term nutrient. Because most of the times nutrient refers to nutrition, and this may be mis-interpreted as an environmental component of milk. Milk composition is influenced by environmental factors surely, but it is also influenced by the cow’s own genetics. Therefore, the milk composition field does not use the term nutrient. At line 368, just say the colostrum composition differs from the composition of early lactation milk. At other lines, use components instead of nutrients. It will be clear to the reader, and to the milk field.

Another issue is the repetition of results in the Discussion section. As previously suggested, consider combining Results and Discussion section.

At line 381, what was the correlation? Please specify.

Reviewer 2 Report

Although the presented topic is interesting (relatively little is known about elephants' milk composition) in my opinion the study have a serious methodological mistakes which do not let me to recommend the evaluated paper for publication in Animals:

  1. The number of tested animals is low but the main problem is that even this small group is totally not homogenous. Some of investigated elephants were milked through the whole lactation, some of them just for several months etc. Some of tests were conducted for just two individuals. 
  2. Animals derived from several different locations, the feeding management was not standarized which is crucial in experiments regarding milk composition. 
  3. There are no information about lactation number of particular elephants - this may also influence the final results.
  4. Authors did not conducted any statistical analysis to support their findings. This is related probably with the mentioned above problems with experimental design (small and not homogenous animal cohort).
  5. Manuscript language should be extensively revised. Some parts are extremaly difficult to follow by the authors' thoughts. Abstract should contains a few sentences of introduction, brief information about material and methods, final results and conclusions.

Reviewer 3 Report

paper 3.2020
